# Learning Local Descriptor for Comparing Renders with Real Images

**Pamir Ghimire [1]**, **Igor Jovančević [1,*]** and **Jean-José Orteu [2]**

1    DIOTASOFT, 31670 Labège, France; pamirghimire@gmail.com
2    Institut Clément Ader (ICA), Université de Toulouse, CNRS, IMT Mines Albi, INSA, UPS, ISAE,
     Campus Jarlard, 81013 Albi, France; jean-jose.orteu@mines-albi.fr
*    Correspondence: igorjovan@gmail.com

**Abstract:** We present a method to train a deep-network-based feature descriptor to calculate discriminative local descriptions from renders and corresponding real images with similar geometry. We are interested in using such descriptors for automatic industrial visual inspection whereby the inspection camera has been coarsely localized with respect to a relatively large mechanical assembly and presence of certain components needs to be checked compared to the reference computer-aided design model (CAD). We aim to perform the task by comparing the real inspection image with the render of textureless 3D CAD using the learned descriptors. The descriptor was trained to capture geometric features while staying invariant to image domain. Patch pairs for training the descriptor were extracted in a semisupervised manner from a small data set of 100 pairs of real images and corresponding renders that were manually finely registered starting from a relatively coarse localization of the inspection camera. Due to the small size of the training data set, the descriptor network was initialized with weights from classification training on ImageNet. A two-step training is proposed for addressing the problem of domain adaptation. The first, "bootstrapping", is a classification training to obtain good initial weights for second training step, triplet-loss training, that provides weights for extracting the discriminative features comparable using $l_2$ distance. The descriptor was tested for comparing renders and real images through two approaches: finding local correspondences between the images through nearest neighbor matching and transforming the images into Bag of Visual Words (BoVW) histograms. We observed that learning a robust cross-domain descriptor is feasible, even with a small data set, and such features might be of interest for CAD-based inspection of mechanical assemblies, and related applications such as tracking or finely registered augmented reality. To the best of our knowledge, this is the first work that reports learning local descriptors for comparing renders with real inspection images.

**Keywords:** deep convolutinal neural networks; learned feature descriptor; 3D CAD; nonphotorealistic rendering

## 1. Introduction

In industrial visual inspection based on 3D computer-aided design (CAD) models, we want to check whether produced mechanical assemblies actually conform with CAD specification.

In this paper, we are focused on one of the largest families of inspection problems, usually called the presence–absence problem. Namely, we are aiming to verify the presence of the right part at the right place in a complex mechanical assembly. To perform this inspection task, we are interested in using images from passive 2D RGB cameras since they are cheap and convey enough information to make many inspection decisions. To check acquired images for conformity with CAD, we propose to compare them with simple renders of the CAD with similar viewpoints using local keypoint features described with our learned descriptor. This comparison should tell us if the part we are looking for based

on the CAD is present or absent in the real assembly given its inspection image. We have CAD models that lack material related information (color and texture) and so we work with simple renders that reveal geometry of the CAD, specifically, changes in normals and depths of faces of the CAD model. Our CAD models also lack deformable elements such as wires and plastic caps that are present in the real assemblies, and occlude the elements we want to check. In tackling this challenging setting, we assume that the inspection camera is calibrated and coarsely localized with respect to the inspected assembly, hence providing us approximate camera viewpoint (pose + intrinsics) for each inspection image. For this, we rely on an in-house developed localization module based on 2D/3D alignment. The CAD can thus be rendered with a viewpoint similar to the real image.

In order to check whether certain components, henceforth inspection elements, are mounted on the assembly correctly i.e., at expected positions and with expected orientations as specified in the CAD, we propose to compare learned local features extracted from real images with those from corresponding renders at interest points such as FAST corners [1].

This is an object detection approach using learned features to check whether an element is present in the real image the way it is in the CAD render. We posit that such learned features are more informative and hence more discriminative than simple 2D features such as contours and edges. We want to use such features because the images we want to inspect are challenging, while our reference CAD models are simple and incomplete, as described before. The inspection elements that interest us are industrial parts such as supports, clamps, etc. that have rigid and regular structures compared to natural everyday objects that have more nonrigid geometries. Because of this, the 2D features can be attractive choices for comparing a shape in a real image with that in the render [2,3]. However, the 2D features do not capture view-point dependent information since it is possible to obtain, for example, same contours from different viewpoints due to object symmetries. Comparing edges in render with those in a real image can also be misleading since edges might result from textures, shadows, deformations, surrounding objects, etc. Such features are hence not informative and discriminative since they capture only simple geometric characteristics. In contrast, $2\frac{1}{2}D$ features, which are features computed from image patches centered at interesting points, capture more detailed information that is revealed due to viewpoint and are robust to the problems faced by 2D features as described before [4].

Local features, engineered as well as learned, have been extensively studied for comparing images, finding local correspondences and detecting objects given a template [5–7]. The common denominator among the studied features has been comparison between only real images. Consequently, these features capture local texture information, important in real images, in addition to local geometric information. Consideration of texture information is problematic when the CAD models lack material specifications restricting renders to plain renders. We are thus interested in a feature extractor that can produce descriptors to compare textureless patches from plain renders with textured patches from real images. In other words, we want our feature extractor to be as much as possible invariant to texture and to work even in absence of any texture in the compared images. Additionally, we also want to ignore deformable elements such as wires and plastic caps in real image patches when extracting descriptors since we do not have them in our simple CAD models; we are restricted to low poly models that use simple primitives to convey complex shapes. Given the success of learned features in comparing real image patches, we are interested in training a neural network that can extract features from real and rendered image patches that satisfy the requirements described before.

Due to the unique nature of our problem, we face two main challenges:

- Lack of a data set of corresponding real and rendered image patches
- Domain shift due to the patches coming from different domains

Since ours is the first work, to the best of our knowledge, that tries to learn a local descriptor for comparing renders of simplistic CADs of mechanical assemblies and real inspection images, we are unable to use available data sets of real image patches, like in [8].

We create our own data set using <100 real-rendered image pairs, with an initial coarse registration and a manual fine registration. We then crop the images around interest points detected by the FAST detector [1]. This method for producing a patch-pair data set is our first contribution. The images we used were taken by an industrial camera mounted on a robotic platform that inspected an assembly from a set of pre-planned positions during scheduled inspections. The time for inspection and the time on the imaging platform were both limited, because of which a limited number of real images with approximate viewpoint information were available for us to work with. Since we also produce one render for each real image, this also limited the number of renders. Our second contribution is a two-step approach for training a neural-network feature descriptor. Because our patches come from two different domains (real and rendered) and are limited in number, we can not train a neural-network descriptor from scratch. Direct minimization of a metric loss by initializing the network with weights from training on real images data set, such as ImageNet [9], is not enough to deal with domain gap because such initialization causes features extracted from patches from different domains to be different. A two-step training approach that first removes distinction between domains before learning discriminative features allows the descriptor network to converge despite small size of our data set. We show that the first step of our approach explicitly addresses the problem of domain shift.

## 2. Related Literature

Local features that describe local geometries can be more discriminative than 2D features that capture information about only segments and corners. Local features have also shown to be useful for a wide variety of vision tasks such as image matching, object detection, etc., and so are more generic than 2D features. Recently, local features learned using a deep network have been shown to be more discriminative than engineered ones [6,10,11]. These features are mostly learned using a data set of natural patches [8] on which either a classification loss or a metric loss is minimized on matching and nonmatching patch pairs. Two metric losses are commonly used, the contrastive loss [12] and the triplet loss [13]. Available patch-pair data sets, in addition to containing only real patches, are also created from images with wide baseline and are meant for training descriptors for such scenarios.

In recent literature, many methods have been proposed for generating and using synthetic data (renders) for performing different tasks such as classification, segmentation, object detection, etc. on real images. In most works, the models that are used have accurate descriptions of color and texture as they appear on the real object [14,15] and allow the learned object detector to make decisions on the real image based on these cues. In some works, however, use of color and texture cues is not prioritized. Ros et al. [16] for example try to learn dense segmentation pipeline for real urban scenes using "photorealistic" renders that do not have elements that correspond exactly in color or texture to the real urban objects. Furthermore, it is shown in [14] that an object detector can be learned using randomly colored and textured object models such that the learned detector is invariant to these cues, and sensitive only to object geometry. These works suggest that it should be possible to train a descriptor that uses only geometric cues.

Although the ease of rendering abundant labeled data makes rendered images attractive for deep learning tasks, the key challenge of domain shift or domain adaptation makes this task difficult [17]. This problem arises because geometric and photometric details available in real images are hard to replicate in renders and real and virtual cameras are different sensors [16]. Prior works have dealt with this problem by training jointly on renders and real images one way or another. In [16], deep networks are trained to predict same object labels for instances in renders and real images, and it is observed that networks learn to extract features that work in both domains for the trained task. In [18], minimization of the domain gap is prioritized explicitly by backpropagating the negative of gradient of domain label prediction loss through a gradient reversal layer in addition to backpropagating the gradient for the main task of classification. The authors of [19] learned to map features extracted from real images to the feature space of renders and

supply the mapped features to a different network to predict pose of a target object. This allows the pose predictor to be trained even exclusively on renders, while the mapping network is trained by creating a set of renders for a set of real images where the pose of the target object in the renders is the same as that in the real images. The authors of [20] used representations learned from a large labeled data set of images of one modality as supervisory signals for learning representations for images of a different modality for which labeled data sets are not available. Representations from RGB images are used to supervise learning on paired depth and optical flow images, paired images being images of the same scene in a different modality, by training networks on these images that extract features similar to the RGB network at some specified network depth. The authors of [21] trained an end-to-end CNN for 2D–3D exemplar detection by compositing rendered views of textured CAD models on real images, and maximising the similarity between features extracted from composites and real images with the same poses of the target objects. This approach is similar to the feature mapping approach proposed in [19]. The authors of [22] build templates automatically from 3D models that can be used for object detection and pose estimation of textureless objects in real images without adaptation. The templates for detection are created by using color gradients at silhouette boundaries of renders of target objects and the ones for pose estimation are created by using surface normals in the interiors as available in the 3D models.

For learning to associate renders of 3D CAD models with their real images, data sets that contain images with exact pose information of the models are important. IKEA Objects [23] is one such data set that contains textureless 3D models of furniture such as cupboards, tables, etc., along with real images that are annotated with exact poses of these objects. Pix3D [24] is another large data set of image-shape pairs (shape here meaning textureless 3D CAD models) representing a wide variety of objects such as chairs, candle holders, etc., that accurately match the objects in available real images and have precise pose annotations. Similarly, in [25], it is an RGBD data set of textureless objects that resemble industrial parts and contains natural scenes with pose annotations where these objects are placed against a background, multiple objects at a time, and with some overlaps. The authors of [22] proposed their own data set with models of 15 textureless household objects and more than 1000 images with pose annotations. The data sets mentioned here are primarily concerned with household or everyday items, and those that do contain industrial parts contain images of relatively easy and well-organized setups of the parts. Therefore, we created our own data set.

## 3. Methodology

There are two main parts to our method for training a cross-domain descriptor. The first part is generation of patch-pairs from real images and renders of CAD models, both from the same viewpoints. The second part is training a deep network initialized with weights from classification training on a large data set (ImageNet [9], weights available freely on the Internet (ImageNet weights for VGG https://github.com/machrisaa/tensorflow-vgg accessed on 1 February 2018)) using a two-step approach. Similar to the literature mentioned before, we also train jointly on renders and real images. The network is initialized with weights obtained from training on real images in the Imagenet data set. Then, through a two-step approach, the weights are adapted so that our network extracts similar features from real and rendered image patches that correspond to the same area of the assembly.

In the following two Sections 3.1 and 3.2, we will present our data set generation process. We are interested in patch pairs that come from renders of industrial assemblies where inspection elements might be components such as screws and brackets. Furthermore, we want to teach our network to handle real images with challenging lighting and occlusion due to wires, plastic caps and other elements that are not available to us in CAD. The second step, training a deep network, is detailed in Section 3.3.

### 3.1. Producing Simple Renders

We used simple shaders for producing renders that convey changes in face normals and face depths with respect to the optical axis of the camera. No lighting was considered in producing these renders so that intensities in the render were only due to geometry of the CADs. In addition, the CADs of the inspection element were rendered to be brighter than the CAD of the assembly. These choices ensured that a detector would find an interest point only at those locations in the render that corresponded to locations with depth or surface orientation discontinuities in the CAD, or discontinuities in element identity. The simple shader has been illustrated in Figure 1. We used Unity3D [26] for creating our synthetic data generation pipeline.

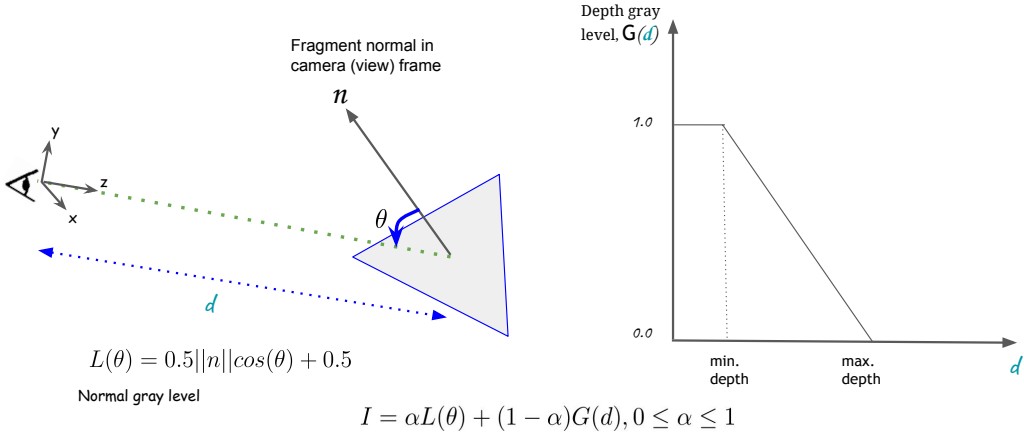

**Figure 1.** Simple shading for fast rendering based on fragment normal $n$ and fragment depth $d$. $L(\theta)$ is the component of fragment's intensity in render due orientation $\theta$ of $n$ relative to render camera's optical axis, and $G(\theta)$ that due to its depth. Final intensity in render, $I$, is an affine combination of the two, with manually chosen $\alpha$, which is the same for all fragments. $I$ is also varied based on whether the fragment belongs to CAD of inspection element or that of assembly on which it is mounted.

### 3.2. Creation of Patch-Pair Data Set

We intend to train a descriptor. We are inspired by the approaches in [6,7,10] and we formulate a similar one with adaptations to address our particular problem. For this, we need a data set of corresponding patches that come from renders and real images. Since there isn't an existing one to the best of our knowledge, we create such a data set by first producing for each real image a simple render using the available viewpoint, and CADs of inspection elements and those of assemblies they are mounted on. Second, we finely register each render with its corresponding real image by manually selecting a set of corresponding points between the two images and estimating a homography that registers the render with the real image. This is necessary because of the inaccuracy in our in-house localization module, as a result of which the renders are not perfectly registered with the real images. Finally, to create the patch-pair data set, we crop each pair of real-rendered images around same points that are detected by the FAST detector in the render. The patch-pairs are created only using locations that are deemed interesting in the render since we want patches that capture interesting geometry in the plain CADs and their corresponding manifestations in real images, not patches that are found interesting in real images by themselves.

We also produce patches by cropping renders and real images around most salient points detected by the FAST detector exclusively in the real images, to produce 'texture' patches. The real ones of these patches contribute to a reservoir of texture patches when training a descriptor by minimizing triplet loss and are always labeled as negatives in the triplets (see Section 3.3.2). Our entire data set generation procedure is illustrated in Figure 2.

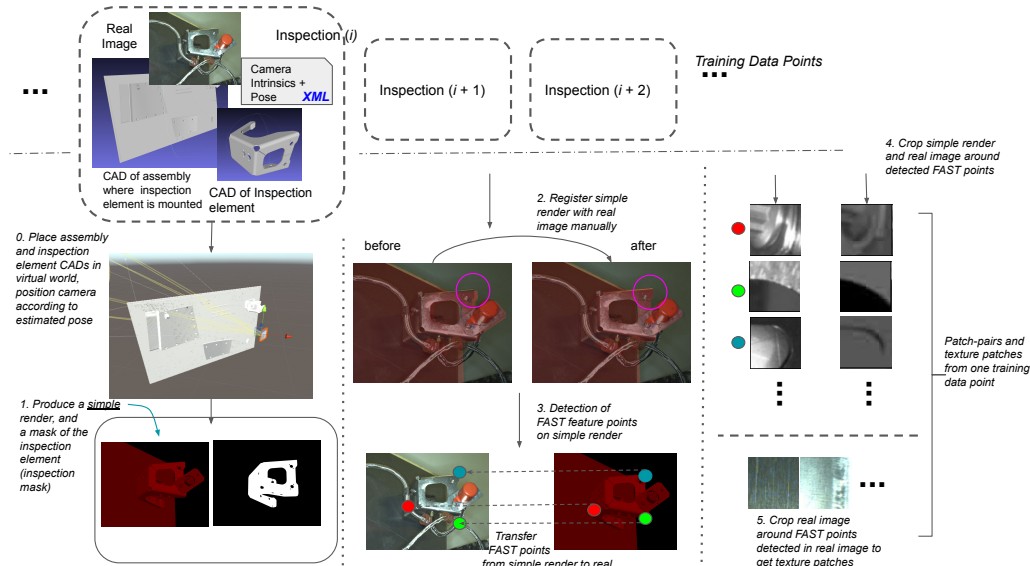

**Figure 2.** Proposed method for creating a patch-pair data set given CADs of inspection elements and their environments along with real images with tracked viewpoint. The simple renders (in red) are manually registered with the corresponding real image. Effect of registration is visible inside the pink circle (images after step 2). From each registered image pair, patch pairs are produced by cropping around FAST points detected in the render.

There are 2 main challenges in training with the patch-pair data set produced as described in Section 3.2:

- Small number of patch pairs due to less than 100 image pairs available for producing the training data set, and need for large patch sizes for discriminative context, due to simplistic CADs
- Domain shift due to patches coming from synthetic and real domains

These two problems demanded a two-step approach, much like the one in [13], with our modifications in the "bootstrapping" stage. The two-step method is in contrast with related works like [6,7,10] that directly minimize a discriminative or metric loss on matching and nonmatching patches from a data set. We also tried this approach but this lead to no convergence, possibly because of the two reasons mentioned before. However, in face recognition literature, a similar two-step method was reported, although with different strategies for "bootstrapping" and without identifying the stage explicitly as meant for addressing domain-shift [13]. Our entire training process will be detailed in the following two Sections 3.3.1 and 3.3.2 and is sumarized in Figure 3.

### 3.3. Two-Step Training for Feature Extractor

#### 3.3.1. Bootstrapping

We started with the VGG16 deep architecture with ImageNet weights for classification training, as detailed in [27] (Table 2, Column D). The filters in the first convolutional layer were modified from $3 \times 3 \times 3$ to $3 \times 3 \times 1$ and their weights initialized by averaging the 3D filters along the 3rd dimension [28]. The 1000-way linear+softmax layer was replaced with 2-way linear+softmax layer. The 2 fully connected layers following the convolutional layers were initialized from scratch using Xavier initialization [28] (Figure 3, step 1). In the original architecture, the fully connected layers have 4096 neurons each. For us, they have $L$ neurons each, with $L = 1024$ when patches are of size $128 \times 128$ and $L = 4096$ when they are $224 \times 224$.

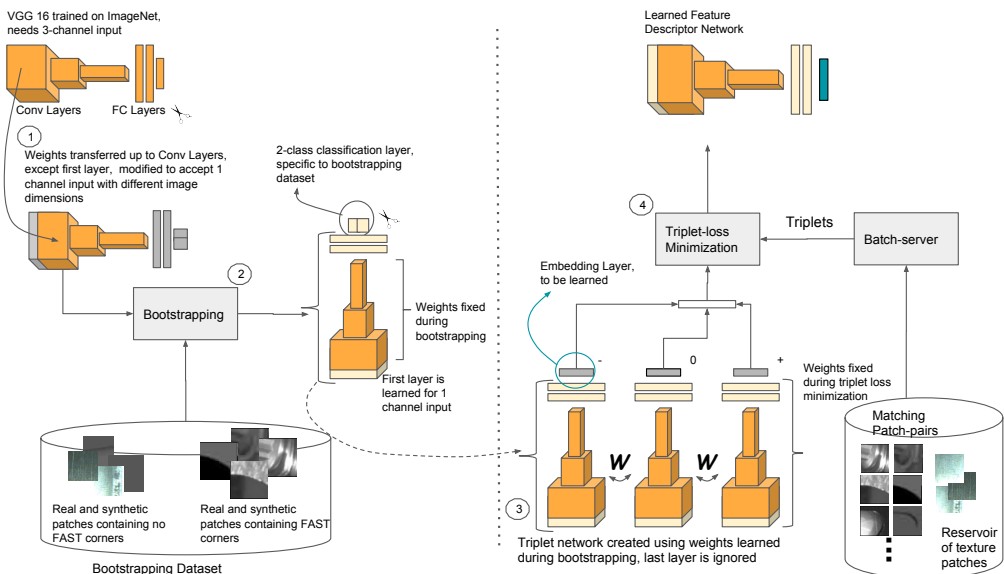

**Figure 3.** Proposed two-step method for training a deep descriptor using a data set of patches created as described in Figure 2. The first stage (left of the vertical divider), bootstrapping, produces weights such that distinction between domains is removed and the second stage (right of the divider), triplet-loss training, trains an embedding layer on top of the weights learned previously that produces discriminative features.

For bootstrapping, the patch-pair data set described before was organized into 2 classes, one containing meaningful geometric information and the other not (texture patches). Both classes contained patches from both real and synthetic images. A cross-entropy loss (Equation (1)) was minimized to classify these patches. All layers except the first convolutional and the last fully connected layers were "frozen" during this stage (Figure 3, step 2). Training was performed for two epochs with minibatch gradient descent using batches of size 128, initial learning rate was 0.005, and dropout was used after fully connected layers for regularization with $p = 0.5$. The softmax-cross-entropy loss (E) that we minimized is:

$$E = \frac{1}{n} \sum_i [y_i log(\hat{y_i}) + (1 - y_i) log(1 - \hat{y_i})] \tag{1}$$

where $n$ is the total number of training examples, $y_i \in \{0, 1\}$ is the patch class label of the training example $i$ ($y_i = 0$ means the patch contains interesting geometry, and $y_i = 1$ means that it doesn't, and hence, is a texture patch) and $\hat{y_i} = \frac{e^{x_i}}{e^{x_0} + e^{x_1}}$ is the predicted softmax probability for example $i$, $x_0$ and $x_1$ being the predicted probabilities for the two patch classes.

### 3.3.2. Triplet Loss Training

The weights learned from bootstrapping were retained except for the last 2-way linear+softmax layer. We denote the retained architecture with $\phi$. It maps an input patch $P \in \mathbb{R}^M$ to $\mathbb{R}^D$; $\phi(P) \in \mathbb{R}^D$. To $\phi$, we appended an $l_2$ normalization layer and a linear layer $W' \in \mathbb{R}^{L \times D}$, $L << D$, which together implement $e_P = W'\phi(P)/||\phi(P)||_2$. $W'$ is an affine transformation layer without any bias, since bias would be canceled when minimizing triplet loss detailed in Equation (2). The thus augmented $\phi$ is replicated three times to create a triplet network [13,29] and triplet loss [13] is minimized for batches of triplets (anchor (a), positive (p), negative (n)) produced by a batch server. In each triplet, an anchor is a negative or a positive example. The weights are shared among all the towers in the triplet network, so they receive the same weight updates during training.

The rationale behind triplet-loss minimization is to learn weights such that distances between matching patches is minimized while at the same time, that between nonmatching patches is maximized.

$$E(W') = \sum_{(a,p,n) \in T} max\{0, \alpha - d_{an} + d_{ap}\} \qquad (2)$$

Here, $d_{an} = ||e_a - e_n||_2$, $d_{ap} = ||e_a - e_p||_2$, and $T$ is the set of triplets in each training mini-batch. Embeddings extracted from patches through a forward pass are denoted by $e_x$. $\alpha > 0$ is the margin parameter that forces a desired minimum difference between average distances between matching and nonmatching patches. The method and notations followed in this section are similar to those in [13].

During triplet-loss training, only the "embedding layer" $W'$ is updated, the rest of the layers are frozen. Thus, it is only $W'$ that learns a discriminative projection of features extracted by $\phi$ (Steps 3 and 4 in Figure 3), learned during bootstrapping. The architecture $\phi$ together with the $l_2$ normalization and $W'$ layers is the learned cross-domain feature extractor. The extracted features are of dimension $D$, with $D = 512$ when patches are of size $128 \times 128$ and $D = 1024$ when they are of size $224 \times 224$. Of the $D's$ we experimented with, these were the best values.

During triplet-loss training, from each batch, we pick "hard" triplets, i.e., triplets for which $d_{an} < d_{ap}$ so that the network can learn from minimizing the triplet loss computed from them. To perform hard-negative mining, we filter a batch of triplets produced by the triplet batch server by passing them through the network once before training for getting their embeddings from which three pairwise distances are computed. We also perform "in triplet hard-negative mining", introduced by [10], by swapping anchor and positive patches if $d_{an} > d_{pn}$. This improves triplet-loss training by making triplets even harder. We use the Adam optimizer with an initial learning rate of 0.005, $\beta_1$ 0.9, $\beta_2$ 0.999 and $\epsilon$ $10^{-08}$. Regularization was performed via dropout added before the embedding layer.

When picking negative real patches for a geometrically interesting patch from a render, batch server has a choice of picking either a geometrically interesting nonmatching real patch or a real texture patch, of which there are thousands in a "reservoir". We undersample the texture patches with ratio of 3:7 to address class imbalance [30]. Small random rotations are also applied to patches on the fly so that the descriptor learns to be robust to rotations [6].

### 3.4. Computing Learned Descriptor at Test-Time

At test time, FAST points (or equivalently some other interest points) are detected independently in real image and the corresponding render. A patch of a fixed size is extracted around each interest point location and the learned descriptor evaluated for it. The features extracted from the images can then be used for nearest neighbor matching (See Figure 4) or converted into histograms of words based on Bag of Visual Words learned from renders and real images using the learned features for comparing the pair for producing an inspection decision.

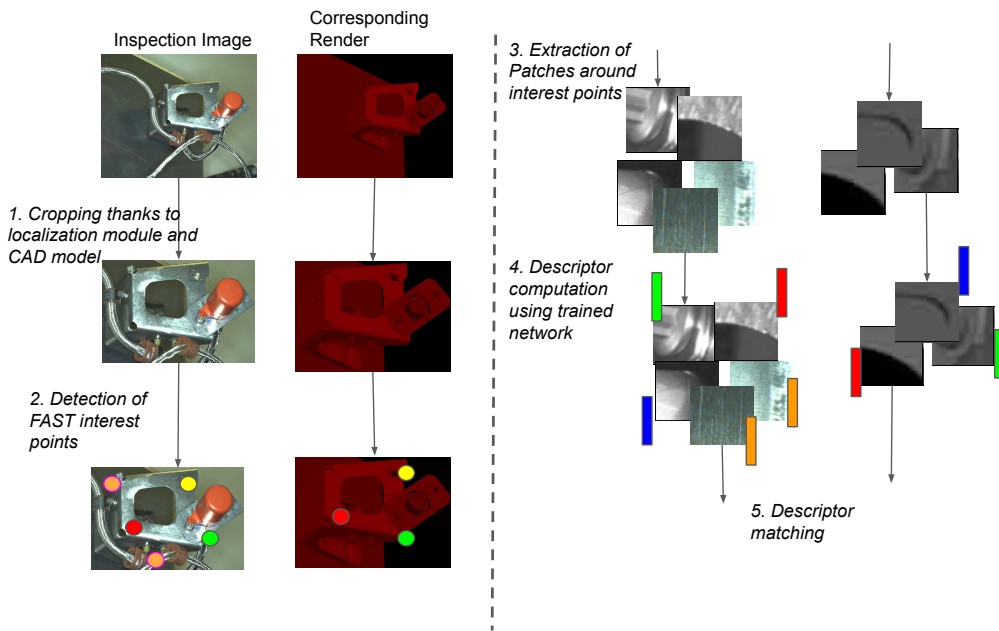

**Figure 4.** Proposed workflow for using the descriptor at test-time for nearest-neighbor one-to-one matching.

## 4. Results

We trained two different networks with patch-pairs of two sizes, $128 \times 128$ and $224 \times 224$, and different values of the margin parameter $\alpha$ (see Equation (2)). Table 1 presents the FPR95 rates, which are false positive rates when true positive rates are 95%, observed for the different descriptors. They were obtained using a test set of 10K patch pairs that were classified as either matching or nonmatching based on whether the Euclidean distance between their embeddings was below a certain threshold. The best score of 13.8 was observed for $224 \times 224$ descriptor with $\alpha = 5.0$. For reference, this metric for SIFT is reported in [10] to be between 26.0 and 30.0 when matching real image patches with real ones.

**Table 1.** FPR95 rates observed for learned descriptors.

| Margin $\alpha$ | $128 \times 128$ Patch Descriptor | $224 \times 224$ Patch Descriptor |
| --- | --- | --- |
| 2.0 | 40.1 | 17.0 |
| 5.0 | 41.2 | 13.8 |
| 10.0 | 37.8 | 17.5 |

Best FPR95 score of 13.8 was obtained for the descriptor that took as input $224 \times 224$ patches and was trained with $\alpha = 5.0$.

### 4.1. Effect of Two-Step Training

We extracted features from a subset of the test patches using the $224 \times 224$ network with weights for best training before and after the bootstrapping stage and after the triplet-loss training stage and performed dimensionality reduction on these features using t-distributed stochastic neighbor embedding (t-SNE) [31]. The feature plots can be seen in Figure 5.

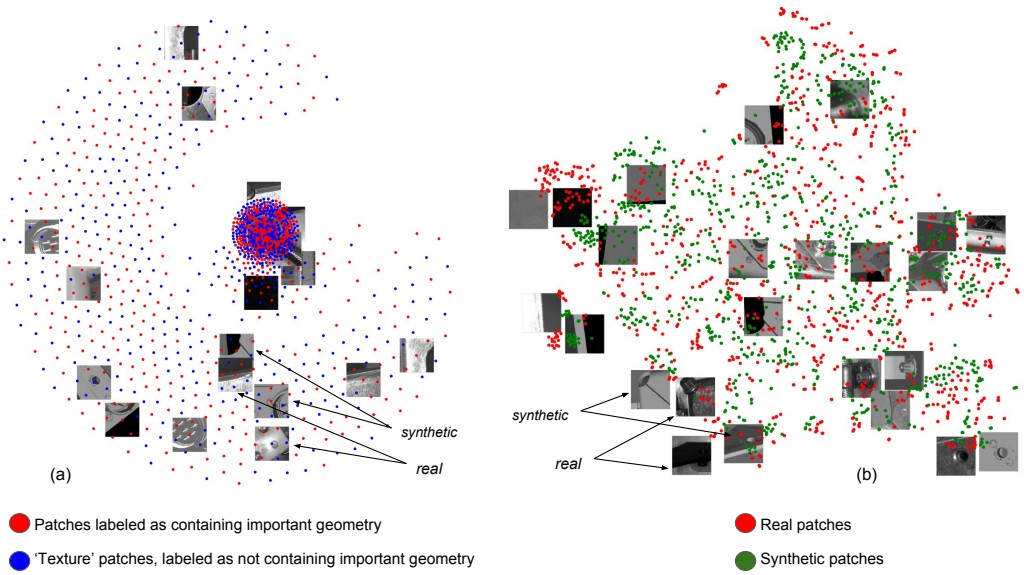

🔴 Patches labeled as containing important geometry

🔵 'Texture' patches, labeled as not containing important geometry

🔴 Real patches

🟢 Synthetic patches

**Figure 5.** Visualization of features extracted from synthetic and real patches after bootstrapping (**a**) and triplet-loss training (**b**) using t-SNE [31]. After bootstrapping, the mappings of real and synthetic patches are close by and intermeshed in the feature space, a good step toward solving domain shift problem, as can be seen from the patches visualized on top of their mapped point features. After triplet-loss training, we see that real and synthetic patches with similar geometry are mapped to nearby locations.

When features are extracted using a network pre-trained on real images without bootstrapping, patches from different domains map to distinct clusters. We observed that such initialization with domain-separation leads to no convergence in the triplet-loss training stage.

This was solved by bootstrapping. We note that after bootstrapping, features extracted from real and synthetic patches are intermeshed since they are based on whether or not the patches contain any texture information. This can be seen from the visualized patches in Figure 5a. After triplet-loss training, the extracted features still remain intermeshed but now are such that those extracted from real and synthetic patches are close together when the patches contain similar geometry, and are far apart if not.

### 4.2. ROC Curves

Figure 6 presents ROC curves, which are plots of true positive vs false positive rates for two different descriptors mentioned in Table 1. Change in $\alpha$ shows little effect on the ROC of the $128 \times 128$ descriptor. $224 \times 224$ descriptors for all values of $\alpha$ perform better than the corresponding $128 \times 128$ descriptor.

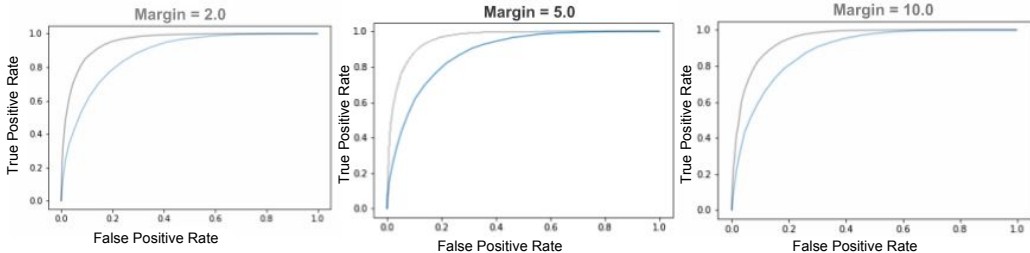

**Figure 6.** ROC curves for $128 \times 128$ (blue) and $224 \times 224$ (gray) learned descriptors for different margin parameters.

In Figure 7, some patch pairs falsely classified as matching or nonmatching by thresholding the FPR95 distance between their learned feature extracted by $224 \times 224$ size network

using the best weights have been presented. In the false-positive as well as false-negative pairs, we see some corresponding geometrical patterns. However, possibly because of lack of sufficient context, the calculated descriptors were not discriminative enough. These suggest that performance might be improved by using still bigger patch sizes.

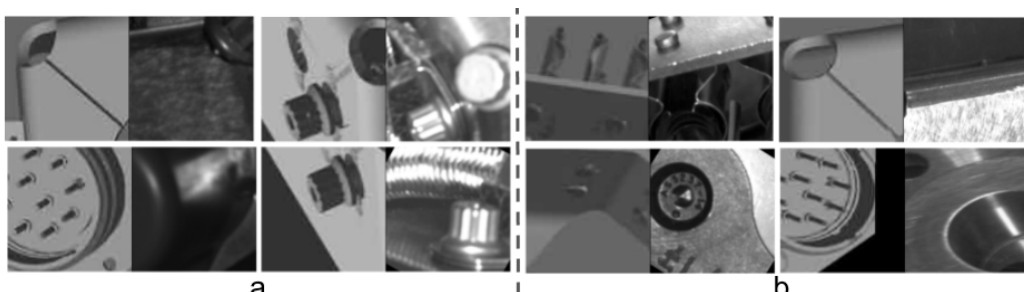

**Figure 7.** Patch-pairs falsely classified as nonmatching (false negatives, four pairs in (**a**)) and matching (false positives, four pairs in (**b**)). In each pair, the left patch comes from a render and the right patch comes from the corresponding real image.

### 4.3. Nearest Neighbor Matching

For nearest neighbor matching between a real image and its corresponding render, we detect interest points in each of the two images using FAST detector. Interest points in adjacent locations are filtered with nonmaximum suppression. At each interest point, we compute a descriptor using our trained deep network as depicted in Figure 4. The descriptors are then matched using a brute-force matcher by comparing their $l_2$ norms. Ambiguous matches are discarded through crosschecking [1].

We present some examples of nearest neighbor matching between renders and real images in Figure 8 . As can be seen, the renders are simplistic and are lacking in parts such as screws and wires that are present in the real images. Results for one-to-one matching were generally poor despite a good FPR95 score. Good ROC does not imply good nearest neighbor matching, as noted in [6].

### 4.4. Bag of Visual Words with Learned Descriptor

Besides nearest neighbor matching for comparing image pairs, we also tested histograms based on BoVW learned from a set of renders and real images by extracting from them ORB features (oriented and rotated BRIEF) [6] and the learned features (LF) around FAST interest points. The two learned BoVW dictionaries both contained 50 words. The choice of 50 words was motivated by work of Mokhtari et al. (2018) [32] who trained an object detector and image classifier on real images similar to the ones we used.

For every test pair of images, the two descriptors, ORB and LF, were calculated at FAST points, and histograms were created using the calculated descriptors and learned dictionaries. For each image, 2 histograms were computed, one based on ORB BoVW and the other on LF BoVW. One pair hence produced 4 histograms, and 2 distances, one between the ORB BoVW histograms and the other between LF BoVW histograms. The distances between all matching test pairs have been shown in Figure 9e. The LF-based distances between histograms were observed to be less erratic than ORB-based ones, as depicted by the histograms and associated entropies in Figure 9a,b.

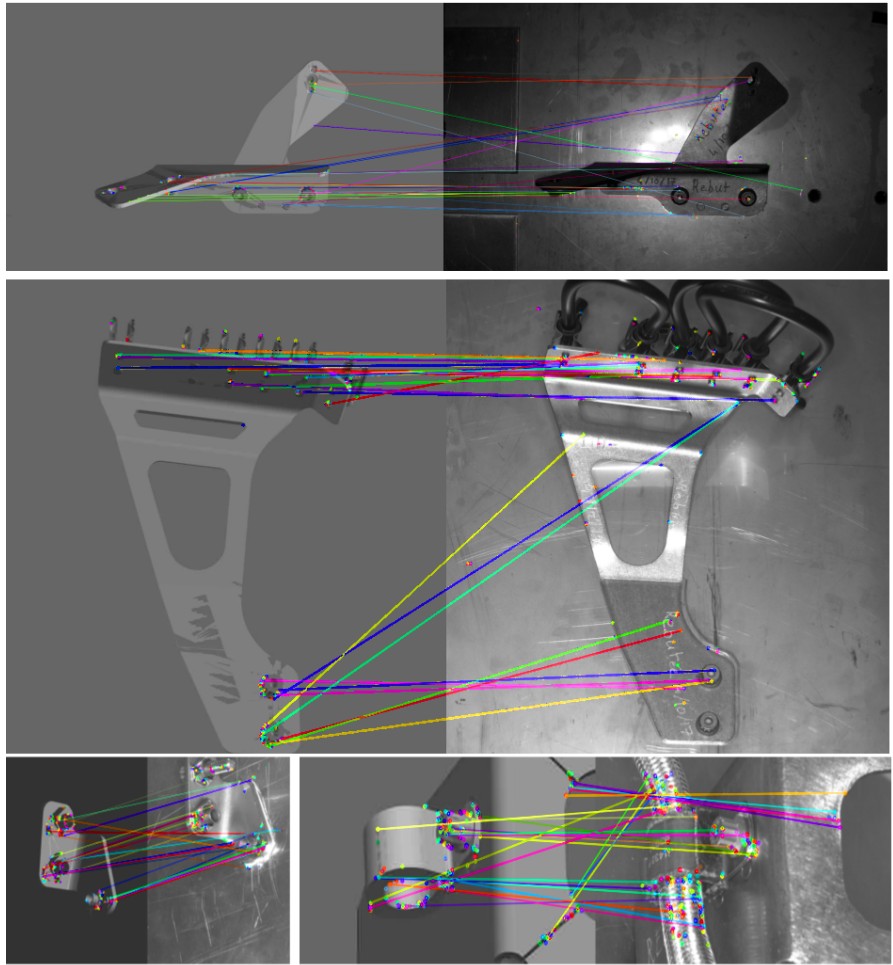

**Figure 8.** Four image pairs showing nearest neighbor matching between renders and real images. In each pair, the left image is a simple render generated for the real image to its right. Only a subset of matches have been shown for clarity.

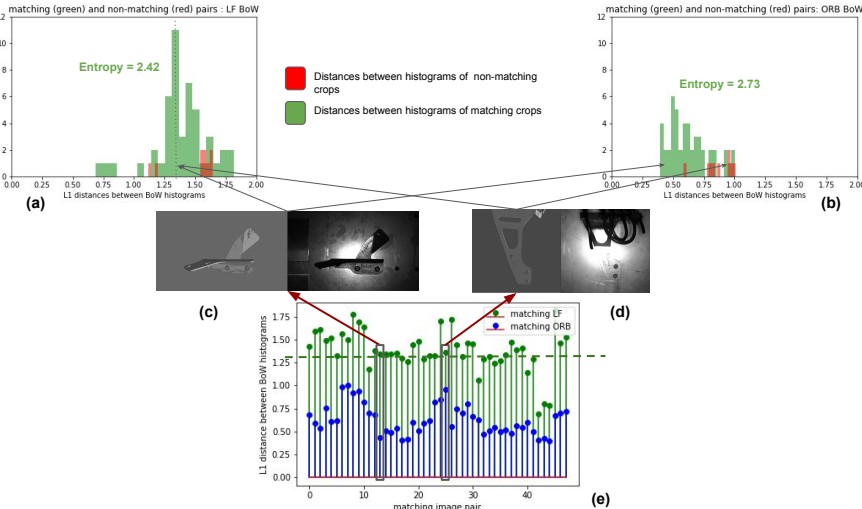

**Figure 9.** Comparison of image pairs using Bag of Visual Words histograms learned from renders and real images using ORB features and the learned features (LF). (**a**,**b**) are distributions of L1 distances between BoVW histograms of matching and non-matching image pairs using LF and ORB features respectively. (**c**,**d**) are matching pairs of rendered and real images. (**e**) is a plot of L1 distances between BoVW histograms of matching image pairs using LF and ORB features.

## 5. Conclusions

We saw that it is possible to learn a feature descriptor to compare simple renders with real images using relatively few image pairs. To do this, we presented a method for producing a data set of patch-pairs, and a method for training the cross-domain descriptor. The learned descriptors showed better performance when larger patches were used, due to simplistic nature of the CADs, i.e., larger patches provided more discriminative context. A two-stage training strategy was necessary for learning the descriptor, where the first stage was for learning initial weights that addressed domain-shift before learning discriminative cross-domain embeddings. The learned descriptors were used for comparing image pairs by nearest neighbor matching and through histograms based on Bag of Visual Words built using the learned features. For nearest-neighbor matching, best FPR95 obtained was 13.8, outperforming SIFT matching for real–real image patches. The learned features can thus be used in a variety of ways, much like other local features.

For future work, we note that although we used a basic shading as described before, a slightly more detailed shading, such as that described by Phong [33], might provide better performance while still enabling fast renders. The deep architecture used was VGG16, which is expensive to evaluate and impractical for real-time inspection. Smaller networks such as the one in [10] need to be explored. The hyperparameters for training the network may be chosen using Bayesian methods [34,35]. The FAST detector was used for sake of dense detections, but its interest points are not best suited for comparing real and synthetic images. Detectors such as LIFT, SIFT, or MSER [5] need to be tested. A detector could also be learned like in LIFT [36]. Instead of matching descriptors one-to-one, matching triplets of descriptors might be an effective alternative, since correspondence between a simplistic CAD and real assembly is not one-to-one. Finally, realistically colored and textured renders with different configurations of the CADs can be explored for augmenting the set of real images.

**Author Contributions:** Conceptualization, P.G., I.J., and J.-J.O. ; methodology, P.G.; software, P.G. and I.J.; validation, P.G. and I.J.; investigation, P.G., J.-J.O., and I.J.; resources, J.-J.O. and I.J.; data curation, I.J.; writing—original draft preparation, P.G.; writing—review and editing, P.G., I.J., and J.-J.O.; supervision, J.-J.O. and I.J.; project administration, J.-J.O.; funding acquisition, J.-J.O. All authors have read and agreed to the published version of the manuscript.

**Funding:** This research was funded by IMT Mines Albi and the company Diotasoft. The APC was funded by IMT Mines Albi.

**Institutional Review Board Statement:** Not applicable.

**Informed Consent Statement:** Not applicable.

**Data Availability Statement:** All the data used in this work are highly sensitive industrial clients data, hence can not be shared due to their confidential nature.

**Acknowledgments:** We would like to thank the whole team at Diotasoft Toulouse office, namely, Yannick Porto, Nour Islam Mokhtari, Benoît Dolives, Hamdi Ben Abdallah, and Ludovic Brèthes, for their support, guidance, and feedback throughout the span of this work.

**Conflicts of Interest:** The authors declare no conflict of interest.

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
