# Peer review of "Learning Local Descriptor for Comparing Renders with Real Images"

_applsci, doi:10.3390/app11083301_

Round 1

Reviewer 1 Report

This paper presents a method to train a deep network-based feature descriptor to calculate discriminative local descriptions from renders and corresponding real images with similar geometry. The authors utilized the established descriptors for automatic industrial visual inspection. This paper contains enough material and experimental results to be publishable.

The experimental results shown are quite impressive and I like the style of presentation and the paper is well organized.

So I recommend the paper.

Author Response

Our answers in the Response, as well as all the changes in .tex fila have been highlighted in blue.

Reviewer 2 Report

In this article, the authors present a method for learning feature descriptors using deep learning algorithms as feature extractors to compute discriminative local descriptions for textureless patches from plain renders with texture patches from real images.  The model has been trained to capture geometric features with invariant images. In addition, patch pairs were extracted using the semi-supervised method.

This article provides two main contributions: first, it provides a synthetic dataset with initial coarse registration and manual fine generation; Second, it uses a two-step approach for training a deep neural network feature descriptor by first removing the distinction between domains before learning the discriminative features, which allow the descriptor network to converge even with small size dataset.

 Experimental validation of the proposed solution shows that learning a robust cross-domain descriptor is feasible with a small dataset, which could be of interest to CAD-based inspection of mechanical assemblies.

However, I have the following concerns about the article.

  • Other than the challenges in training the patch-pair datasets, authors need to clearly define the problem being addressed even though the contributions being made have been articulated. The main problem being addressed is not clearly defined.
  • The article is a bit hard to follow due to language issues. There is a need for improvement in the presentation.
  • Synthetic data has been generated corresponding to real and rendered image patches because there is no real-world dataset to be used, however, why have you used a small dataset?
  • On line 356, the authors claim that the BoVW contains 50 words, how did you decide on how many words each bag should contain?
  • The authors need to provide more detail on the nearest neighbor matching method.
  • The quality of figure 2 should be improved. It’s difficult to read the text labels.

Author Response

(The authors gave the same response as above.)
